# RotDCF: Decomposition of Convolutional Filters for Rotation-Equivariant Deep Networks

**Xiuyuan Cheng, Qiang Qiu, Robert Calderbank & Guillermo Sapiro**
Department of Mathematics, Department of Electrical & Computer Engineering
Duke University
Durham, NC 27708, USA
`{xiuyuan.cheng,qiang.qiu,robert.calderbank,guillermo.sapiro}@duke.edu`

## Abstract

Explicit encoding of group actions in deep features makes it possible for convolutional neural networks (CNNs) to handle global deformations of images, which is critical to success in many vision tasks. This paper proposes to decompose the convolutional filters over joint steerable bases across the space and the group geometry simultaneously, namely a rotation-equivariant CNN with decomposed convolutional filters (RotDCF). This decomposition facilitates computing the joint convolution, which is proved to be necessary for the group equivariance. It significantly reduces the model size and computational complexity while preserving performance, and truncation of the bases expansion serves implicitly to regularize the filters. On datasets involving in-plane and out-of-plane object rotations, RotDCF deep features demonstrate greater robustness and interpretability than regular CNNs. The stability of the equivariant representation to input variations is also proved theoretically. The RotDCF framework can be extended to groups other than rotations, providing a general approach which achieves both group equivariance and representation stability at a reduced model size.

## 1 Introduction

While deep convolutional neural networks (CNN) have been widely used in computer vision and image processing applications, they are not designed to handle large group actions like rotations, which degrade the performance of CNN in many tasks (Cheng et al., 2016; Hallman & Fowlkes, 2015; Jaderberg et al., 2015b; Laptev et al., 2016; Maninis et al., 2016). The regular convolutional layer is equivariant to input translations, but not other group actions. An indirect way to encode group information into the deep representation is to conduct generalized convolutions across the group as well, as in Cohen & Welling (2016a). In theory, this approach can guarantee the group equivariance of the learned representations, which provides better interpretability and regularity as well as the capability of estimating the group action in localization, boundary detection, etc. For the important case of 2D rotations, group-equivariant CNNs have been constructed in several recent works, e.g., (Weiler et al., 2017), Harmonic Net (Worrall et al., 2017) and Oriented Response Net (Zhou et al., 2017). In such networks, the layer-wise output has an extra index representing the group element (c.f. Table 1), and consequently, the convolution must be across the space and the group jointly (proved in Section 3.1). This typically incurs a significant increase in the number of parameters and computational load, even with the adoption of steerable filters (Freeman et al., 1991; Weiler et al., 2017; Worrall et al., 2017). In parallel, low-rank factorized filters have been proposed for sparse coding as well as the compression and regularization of deep networks. In particular, Qiu et al. (2018) showed that decomposing filters under non-adaptive bases can be an effective way to reduce the model size of CNNs without sacrificing performance. However, these approaches do not directly apply to be group-equivariant. We review these connections in more detail in Section 1.1.

This paper proposes a truncated bases decomposition of the filters in group-equivariant CNNs, which we call the rotation-equivariant CNN with decomposed convolutional filters (RotDCF). Since we need a joint convolution over $\mathbb{R}^2$ and $SO(2)$, the bases are also joint across the two geometrical domains, c.f. Figure 1. The benefits of bases decomposition are three-fold: (1) Reduction of the number of parameters and computational complexity of rotation-equivariant CNNs, c.f. Section 2.3;

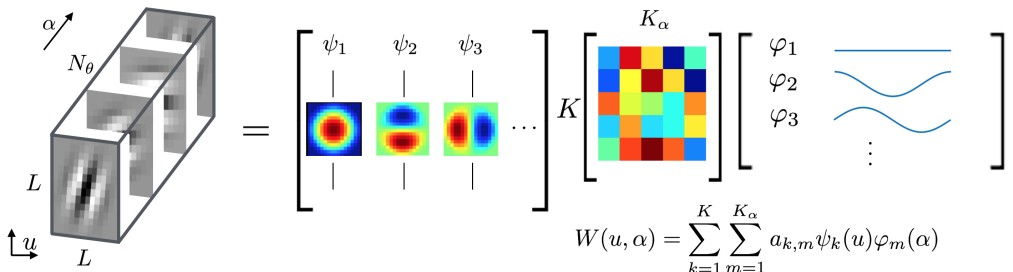

Figure 1: Decomposition of the convolutional filter across the 2D space (variable $u$) and the $SO(2)$ rotation group geometry (variable $\alpha$) simultaneously. The filter is represented as a truncated expansion under the pre-fixed bases $\psi_k(u)\varphi_m(\alpha)$ with adaptive coefficients $a_{k,m}$ learned from data. $\psi_k$ are Fourier-Bessel bases, $\varphi_m$ are Fourier bases, the first 3 of each are shown. The filter has $N_\theta$ group-indexed channels (indexed by $\alpha$) and only one input and output unstructured channel (indexed by $\lambda'$ and $\lambda$ respectively) for simplicity. c.f. Table 1.

(2) Implicit regularization of the convolutional filters, leading to improved robustness of the learned deep representation shown experimentally in Section 4; (3) Theoretical guarantees on stability of the equivariant representation to input deformations, which follow from a generic condition on the filters in the decomposed form, c.f. Section 3.2.

## 1.1 RELATED WORK

**Learning with factorized filters.** In the sparse coding literature, Rubinstein et al. (2010) proposed the factorization of learned dictionaries under another prefixed dictionary. Separable filters were used in Rigamonti et al. (2013) to learn the coding of images. Papyan et al. (2017) interpreted CNN as an iterated convolutional sparse coding machine, and in this view, the factorized filters should correspond to a "dictionary of the dictionary" as in Rubinstein et al. (2010). In the deep learning literature, low-rank factorization of convolutional filters has been previously used to remove redundancy in trained CNNs (Denton et al., 2014; Jaderberg et al., 2014). The compression of deep networks has also been studied in Chen et al. (2015); Han et al. (2016; 2015), SqueezeNet (Iandola et al., 2016), etc., where the low-rank factorization of filters can be utilized. MobileNets (Howard et al., 2017) used depth-wise separable convolutions to obtain significant compression. Tensor decomposition of convolutional layers was used in Lebedev et al. (2014) for CPU speedup. Tai et al. (2015) proposed low-rank-regularized filters and obtained improved classification accuracy with reduced computation. Qiu et al. (2018) studied decomposed-filter CNN with prefixed bases and trainable expansion coefficients, showing that the truncated bases decomposition incurs almost no decrease in classification accuracy while significantly reducing the model size and improving the robustness of the deep features. None of the above networks are group equivariant.

**Group-equivariant deep networks.** The encoding of group information into network representations has been studied extensively. Among earlier works, transforming auto-encoders (Hinton et al., 2011) used a non-convolutional network to learn group-invariant features and compared with hand-crafted ones. Rotation-invariant descriptors were studied in Schmidt & Roth (2012b) with product models, and in Jaderberg et al. (2015a); Kivinen & Williams (2011); Schmidt & Roth (2012a) by estimating the specific image transformation. Gonzalez et al. (2016); Wu et al. (2015) proposed rotating conventional filters to perform rotation-invariant texture and image classification. The joint convolution across space and rotation has been studied in the scattering transform (Oyallon & Mallat, 2015; Sifre & Mallat, 2013). Group-equivariant CNN was considered by Cohen & Welling (2016a), which handled several finite small-order discrete groups on the input image. Rotation-equivariant CNN was later developed in Weiler et al. (2017); Worrall et al. (2017); Zhou et al. (2017) and elsewhere. In particular, steerable filters were used in Cohen & Welling (2016b); Weiler et al. (2017); Worrall et al. (2017). $SO(3)$-equivariant CNN for signals on spheres was studied in Cohen et al. (2018) in a different setting. Overall, the efficiency of equivariant CNNs remains to be improved since the model is typically several times larger than that of a regular CNN. The current paper adopts bases-decomposed filters previously studied in the non-equivariant setting (Qiu et al., 2018), however, the joint convolution scheme in equivariant CNNs is over the two geometries of space and orientation simultaneously and neither the approach nor the analysis there can be directly applied.

## 2 ROTATION-EQUIVARIANT DCF NET

### 2.1 ROTATION-EQUIVARIANT CNN

A rotation-equivariant CNN indexes the channels by the $SO(2)$ group (Weiler et al., 2017; Zhou et al., 2017): The $l$-th layer output is written as $x^{(l)}(u, \alpha, \lambda)$, the position $u \in \mathbb{R}^2$, the rotation $\alpha \in S^1$, and $\lambda \in [M_l]$, $M_l$ being the number of unstructured channel indices. Throughout the paper, $[m]$ stands for the set $\{1, \cdots, m\}$. We denote the group $SO(2)$ also by the circle $S^1$ since the former is parametrized by the rotation angle. The convolutional filter at the $l$-th layer is represented as $W^{(l)}_{\lambda', \lambda}(v, \alpha)$, $\lambda' \in [M_{l-1}]$, $\lambda \in [M_l]$, $v \in \mathbb{R}^2$, $\alpha \in S^1$, except for the 1st layer where there is no indexing of $\alpha$. In practice, $S^1$ is discretized into $N_\theta$ points on $(0, 2\pi)$. We denote the summation over $u$ and $\alpha$ by continuous integration, and the notation $\int_{S^1}(\cdots)d\alpha$ means $\frac{1}{2\pi}\int_0^{2\pi}(\cdots)d\alpha$.

Let the 2D rotation by angle $t$ be denoted by $\Theta_t$, in the 1st layer of the group-invariant CNN,

$$x^{(1)}(u, \alpha, \lambda) = \sigma\left(\sum_{\lambda'=1}^{M_0}\int_{S^1}\int_{\mathbb{R}^2} x^{(0)}(u + v', \lambda')W^{(1)}_{\lambda', \lambda}(\Theta_\alpha v')dv' + b^{(1)}(\lambda)\right). \tag{1}$$

Note that the 1st layer output has $N_\theta$ orientations indexed by $\alpha$, forming a "channel geometry" (Table 1). For $l > 1$, the convolution is jointly over $\mathbb{R}^2$ and $SO(2)$, which takes the form as

$$x^{(l)}(u, \alpha, \lambda) = \sigma\left(\sum_{\lambda'=1}^{M_{l-1}}\int_{S^1}\int_{\mathbb{R}^2} x^{(l-1)}(u + v', \alpha', \lambda')W^{(l)}_{\lambda', \lambda}(\Theta_\alpha v', \alpha' - \alpha)dv'd\alpha' + b^{(l)}(\lambda)\right). \tag{2}$$

The joint convolution over $\mathbb{R}^2$ and $SO(2)$ is both sufficient and necessary to guarantee group-equivariance (Theorem 3.1). While group equivariance is a desirable property, the model size and computation can be increased significantly due to the extra index $\alpha \in [N_\theta]$.

### 2.2 DECOMPOSED FILTERS UNDER STEERABLE BASES

We decompose the filters with respect to $u$ and $\alpha$ simultaneously: Let $\{\psi_k\}_k$ be a set of bases on the unit 2D disk, and $\{\varphi_m\}_m$ be bases on $S^1$. At the $l$-th layer, let $j_l$ be the scale of the filter in $u$, and $\psi_{j,k} = 2^{-2j}\psi_k(2^{-j}u)$ (the filter is supported on the disk of radius $2^{j_l}$). Since we use continuous convolutions, the down-sampling by "pooling" is modeled by the rescaling of the filters in space. The decomposed filters are of the form

$$W^{(1)}_{\lambda', \lambda}(v) = \sum_k a^{(1)}_{\lambda', \lambda}(k)\psi_{j_1,k}(v), \quad W^{(l)}_{\lambda', \lambda}(v, \beta) = \sum_k \sum_m a^{(l)}_{\lambda', \lambda}(k, m)\psi_{j_l,k}(v)\varphi_m(\beta), \, l > 1, \tag{3}$$

which is illustrated in Figure 1 (for $l > 1$). We use Fourier-Bessel (FB) bases for $\{\psi_k\}_k$ and Fourier bases for $\{\varphi_m\}_m$. Specifically, $\varphi_m(\alpha) = e^{im\alpha}$, and FB basis has the expression

$$\psi_k(r, \theta) = c_{m,q}J_m(R_{m,q}r)e^{im\theta}, \quad 0 \le r \le 1, \quad 0 \le \theta \le 2\pi, \quad k = (m, q) \tag{4}$$

where $m$ is the angular frequency, $q$ the radial frequency, $J_m$ the Bessel function ($R_{m,q}$ the $q$-th root of $J_m$) and $c_{m,q}$ a normalizing constant (Abramowitz & Stegun, 1964). Both bases are "steerable", i.e. the operation of rotation is a diagonalized linear transform under both bases. In the complex-valued version, $\psi_k(\Theta_t v) = e^{-im(k)t}\psi_k(v)$, $\varphi_m(\alpha - t) = e^{-imt}\varphi_m(\alpha)$. This means that after the convolutions on $\mathbb{R}^2 \times S^1$ with the bases $\psi_k(v)\varphi_l(\alpha)$ are computed for all $k$ and $l$, both up to certain truncation, the joint convolution (1), (2) with all rotated filters can be calculated by the algebraic manipulation of the expansion coefficients $a^{(l)}_{\lambda', \lambda}(k, m)$, and without any re-computation of the spatial-rotation joint convolution. Standard real-valued versions of the bases $\psi_k$ and $\varphi_m$ in sin's and cos's are used in practice. During training, only the expansion coefficients $a$'s are updated, and the bases are fixed.

| fully-connected layer | regular convolutional layer | CNN with group-indexed channels |
|---|---|---|
| $x^{(l-1)}(\lambda') \to x^{(l)}(\lambda)$ | $x^{(l-1)}(u', \lambda') \to x^{(l)}(u, \lambda)$ | $x^{(l-1)}(u', \alpha', \lambda') \to x^{(l)}(u, \alpha, \lambda)$ |
| $\lambda' \to \lambda$: dense | $u' \to u$: spatial convolution $\lambda' \to \lambda$: dense | $u' \to u, \alpha' \to \alpha$: joint convolution $\lambda' \to \lambda$: dense |

Table 1: Comparison of a fully-connected layer, a regular convolutional layer, and a rotation-equivariant convolutional layer with group-indexed channels.

Apart from the saving of parameters and computation, the bases truncation also regularizes the convolutional filters by discarding the high frequency components. As a result, DCF Net reduces response to those components in the input at all layers, which improves the robustness of the learned feature without affecting recognition performance. The visibly smoother trained filters in RotDCF are shown in Figure A.1, demonstrating the same regularization effects as in Qiu et al. (2018), the latter being without the rotation-equivariant setting. The theoretical properties of RotDCF Net, particularly the representation stability, will be analyzed in Section 3.

### 2.3 NUMBERS OF PARAMETERS AND COMPUTATION FLOPS

**Number of trainable parameters**: In a regular CNN, a convolutional layer of size $L \times L \times M_0' \times M_0$ ($L \times L$ being the patch size) has $L^2 M_0' M_0$ parameters. In an equivariant CNN, a joint convolutional filter is of size $L \times L \times N_\theta \times M' \times M$, so that the number of parameters is $L^2 N_\theta M' M$. In a RotDCF Net, $K$ bases are used in space and $K_\alpha$ bases across the angle $\alpha$, so that the number of parameters is $K K_\alpha M' M$. This gives a reduction of $\frac{K}{L^2} \cdot \frac{K_\alpha}{N_\theta}$ compared to non-bases equivariant CNN. In practice, after switching from a regular CNN to a RotDCF Net, typically $M \leq \frac{1}{2} M_0$ or more due to the adoption of filters in all orientations. The factor $\frac{K}{L^2}$ is usually between $\frac{1}{8}$ and $\frac{1}{3}$ depending on the network and the problem (Qiu et al., 2018). In all the experiments in Section 4, $K_\alpha$ is typically 5, and $N_\theta = 8$ or 16. This means that RotDCF Net achieves a significant parameter reduction from the non-bases equivariant CNN (the factor $\frac{K K_\alpha}{L^2 N_\theta}$ is $\frac{1}{8}$ or more), and even reduces parameters from a regular CNN by a factor of $\frac{1}{2}$ or more.

**Computation in a forward pass**: When the input and output are both $W \times W$ in space, the regular CNN layer needs $2 M_0' M_0 W^2 L^2$ many flops, and a non-bases equivariant convolutional layer needs about $2 M' M W^2 L^2 N_\theta^2$. In contrast, the computation in a RotDCF layer is dominated by a term of $2 M' M W^2 K_\alpha K N_\theta$, which is reduced from the non-bases equivariant network by a factor of $\frac{K}{L^2} \cdot \frac{K_\alpha}{N_\theta}$. Detailed calculations in Appendix A.

In summary, RotDCF Net achieves a reduction of $\frac{K}{L^2} \cdot \frac{K_\alpha}{N_\theta}$ from non-bases equivariant CNNs, in terms of both model size and computation. With typical network architectures, RotDCF Net may be of a smaller model size than regular CNNs. Numbers for specific networks are shown in Section 4.

## 3 THEORETICAL ANALYSIS OF DEEP FEATURES

This section presents two analytical results: (1) Joint convolution (1), (2) is sufficient and actually necessary to obtain rotation equivariance; (2) Stability of the equivariant representation with respect to input variations is proved under generic conditions, which is important in practice since rotations are never perfect.

### 3.1 GROUP-EQUIVARIANT PROPERTY

We consider the change of the $l$-th layer output when the input image undergoes some arbitrary rotation. Let rotation around point $u_0$ by angle $t$ be denoted by $\rho = \rho_{u_0,t}$, i.e. $\rho_{u_0,t} u = u_0 + \Theta_t(u - u_0)$, for any $u \in \mathbb{R}^2$, and the transformed image by $D_\rho x^{(0)}(u, \lambda) = x^{(0)}(\rho_{u_0,t} u, \lambda)$, for any $\lambda \in [M_0]$. We also define the action $T_\rho$ on the $l$-th layer output $x^{(l)}$, $l > 0$, as

$$T_\rho x^{(l)}(u, \alpha, \lambda) = x^{(l)}(\rho_{u_0,t} u, \alpha - t, \lambda), \quad \forall \lambda \in [M_l]. \tag{5}$$

The following theorem, proved in Appendix B, shows that the joint convolution scheme (1), (2) not only produces group-equivariant features at all layers in the sense of

$$x^{(l)}[D_\rho x^{(0)}] = T_\rho x^{(l)}[x^{(0)}], \tag{6}$$

but is necessary for a CNN with $SO(2)$-indexed channels to achieve (6). The sufficiency part is previously shown in Weiler et al. (2017). The necessity of the joint convolution motivates the design and the efforts of reducing the complexity of such models. Note that RotDCF is a type of the channel-indexed CNNs considered in the theorem, so it follows that RotDCF is group-equivariant.

**Theorem 3.1.** *In a CNN with $SO(2)$-indexed channels, let $x^{(l)}[x^{(0)}]$ be the output at the $l$-th layer from input $x^{(0)}(u, \lambda)$. The relation (6) holds for all $l$ if and only if the convolutional layers are given by (1), (2).*

## 3.2 Representation Stability under Input Variations

**Assumptions on the RotDCF layers.** Following Qiu et al. (2018), we make the following generic assumptions on the convolutional layers: First,

(**A1**) Non-expansive sigmoid: $\sigma : \mathbb{R} \to \mathbb{R}$ is non-expansive.

Second, we also need a boundedness assumption on the convolutional filters $W^{(l)}$ for all $l$:

(**A2**) Boundedness of filters: In all layers, $A_l \leq 1$,

where $A_l$ is defined as

$$A_l := \pi \max\{\sup_\lambda \sum_{\lambda'=1}^{M_{l-1}} \|a_{\lambda',\lambda}^{(l)}\|_{\text{FB}}, \sup_{\lambda'} \frac{M_{l-1}}{M_l} \sum_{\lambda=1}^{M_l} \|a_{\lambda',\lambda}^{(l)}\|_{\text{FB}}\}, \tag{7}$$

$$\|a_{\lambda',\lambda}^{(1)}\|_{\text{FB}}^2 = \sum_k \mu_k (a_{\lambda',\lambda}^{(1)}(k))^2, \quad \|a_{\lambda',\lambda}^{(l)}\|_{\text{FB}}^2 = \sum_k \sum_m \mu_k (a_{\lambda',\lambda}^{(l)}(k,m))^2, \quad l > 1, \tag{8}$$

$\mu_k$ being the Dirichlet Laplacian eigenvalues of the unit disk in $\mathbb{R}^2$. Note that (A2) bounds the expansion coefficients, which implies a sequence of boundedness conditions on the convolutional filters in all layers (Proposition B.1), based upon which the stability results below are derived. Since $\mu_k$ typically increases in order, (A2) suggests truncating the series to only include low-frequency $k$ and $m$'s, which is implemented in Section 4. The validity of the boundedness assumption can be qualitatively fulfilled by normalization layers which is standard in practice.

**Non-expansiveness of the network mapping.** Let the $L^2$ norm of $x^{(l)}$ be defined as

$$\|x^{(l)}\|^2 = \frac{1}{M_l} \sum_{\lambda=1}^{M_l} \frac{1}{|\Omega|} \int_{\mathbb{R}^2} \int_{S^1} x^{(l)}(u,\alpha,\lambda)^2 du d\alpha, \quad l \geq 1$$

and $\|x^{(0)}\|^2 = \frac{1}{M_0} \sum_\lambda \frac{1}{|\Omega|} \int_{\mathbb{R}^2} x^{(0)}(u,\lambda)^2 du$. $\Omega$ is the domain on which $x^{(0)}$ is supported, usually $\Omega = [-1,1] \times [-1,1] \subset \mathbb{R}^2$. The following result is proved in Appendix B:

**Proposition 3.2.** *In a RotDCF Net, under (A1), (A2), for all $l$,*

*(a) The mapping of the l-th convolutional layer (including $\sigma$), denoted as $x^{(l)}[x^{(l-1)}]$, is non-expansive, i.e., $\|x^{(l)}[x_1] - x^{(l)}[x_2]\| \leq \|x_1 - x_2\|$ for arbitrary $x_1$ and $x_2$.*

*(b) $\|x_c^{(l)}\| \leq \|x_c^{(l-1)}\|$ for all l, where $x_c^{(l)}(u,\alpha,\lambda) = x^{(l)}(u,\alpha,\lambda) - x_0^{(l)}(\lambda)$ (without index $\alpha$ when l=1) is the centered version of $x^{(l)}$ by removing $x_0^{(l)}$, defined to be the output at the l-th layer from a zero bottom-layer input. As a result, $\|x_c^{(l)}\| \leq \|x_c^{(0)}\| = \|x^{(0)}\|$.*

**Insensitivity to input deformation.** We consider the deformation of the input "module" to a global rotation. Specifically, let the deformed input be of the form $D_\rho \circ D_\tau x^{(0)}$, where $D_\rho$ is as in Section 3.1, $\rho = \rho_{u_0,t}$ being a rigid 2D rotation, and $D_\tau$ is a small deformation in space defined by

$$D_\tau x^{(0)}(u,\lambda) = x^{(0)}(u - \tau(u),\lambda), \quad \forall u \in \mathbb{R}^2, \lambda \in [M_0], \tag{9}$$

with $\tau : \mathbb{R}^2 \to \mathbb{R}^2$ is $C^2$. Following Qiu et al. (2018), we assume the small distortion condition:

(**A3**) Small distortion: $|\nabla \tau|_\infty = \sup_u \|\nabla \tau(u)\| < \frac{1}{5}$, with $\|\cdot\|$ being the operator norm.

The mapping $u \mapsto u - \tau(u)$ is locally invertible, and the constant $\frac{1}{5}$ is chosen for convenience. With $T_\rho$ as defined in (5), the stability result is summarized as

**Theorem 3.3.** *Let $\rho = \rho_{u_0,t}$ be an arbitrary rotation in $\mathbb{R}^2$, around $u_0$ by angle $t$, and let $D_\tau$ be a small deformation. In a RotDCF Net, under (A1), (A2), (A3), $c_1 = 4$, $c_2 = 2$, for any $L$,*

$$\|x^{(L)}[D_\rho \circ D_\tau x^{(0)}] - T_\rho x^{(L)}[x^{(0)}]\| \leq (2c_1 L |\nabla \tau|_\infty + c_2 2^{-j_L} |\tau|_\infty) \|x^{(0)}\|.$$

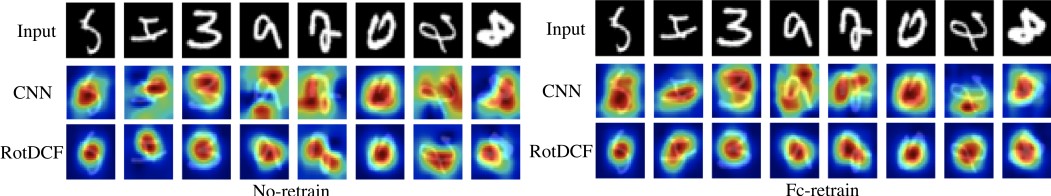

Figure 2: Representative class activation maps (CAM) on testing images in the rotMNIST transfer learning experiment. The heatmap indicates the importance of image regions used in recognizing a digit class. The CNN and RotDCF networks are trained on up-right samples, with no retrain (left) and retraining the fully connected layers respectively (right) before testing. Testing samples are randomly rotated up to 60 degrees. (c.f. Table 2).

| MNIST to rotMNIST MaxRot=30 Degrees | | |
|---|---|---|
| | no-retrain | fc-retrain |
| CNN | 92.61 | 94.71 |
| RotDCF | 96.90 | 98.48 |

| MNIST to rotMNIST MaxRot=60 Degrees | | |
|---|---|---|
| | no-retrain | fc-retrain |
| CNN | 69.61 | 85.90 |
| RotDCF | 82.36 | 97.68 |

Table 2: Test accuracy in the rotMNIST transfer learning experiment. The network is trained on 10K up-right MNIST samples and tested on 50K randomly rotated samples up to the MaxRot degrees.

To prove Theorem 3.3, we firstly establish an approximate equivariant relation for all layers $l$ (Proposition B.2), which can be of independent interest for estimating the image transformations. All the proofs are left to Appendix. Unlike previous stability results for regular CNNs, the above result allows an arbitrary global rotation $\rho$ with respect to which the RotDCF representation is equivariant, apart from a small "residual" distortion $\tau$ whose influence can be bounded. This is also an important result in practice, because most often in recognition tasks the image rotation is not a rigid in-plane one, but is induced by the rotation of the object in 3D space. Thus the actual transformation of the image may be close to a 2D rotation but is not exact. The above result guarantees that in such cases the RotDCF representation undergoes approximately an equivariant action of $T_\rho$, which implies consistency of the learned deep features up to a rotation. The improved stability of RotDCF Net over regular CNNs in this situation is observed experimentally in Section 4.

# 4 EXPERIMENTAL RESULTS

In this section, we experimentally test the performance of RotDCF Nets on object classification and face recognition tasks. The advantage of RotDCF Net is demonstrated via improved recognition accuracy and robustness to rotations of the object, not only with in-plain rotations but with 3D rotations as well. To illustrate the rotation equivariance of the RotDCF deep features, we show that a trained auto-encoder with RotDCF encoder layers is able to reconstruct rotated digit images from "circulated" codes. All codes will be publicly available.

## 4.1 OBJECT CLASSIFICATION

**Non-transfer learning setting**. The **rotMNIST** dataset contains $28 \times 28$ grayscale images of digits from 0 to 9, randomly rotated by an angle uniformly distributed from 0 to $2\pi$ (Cohen & Welling, 2016a). We use 12,000 and 6,000 training samples, and 50,000 testing samples. The baseline net is

| rotMNIST Conv-6, $N_{tr} = 12K$ | | | |
|---|---|---|---|
| | Test Acc. | # param. | Ratio |
| CNN $M$=32 | 96.54 | $6.796\times10^5$ | 1.00 |
| DCF $M$=32, $K$=5 | 96.54 | $1.363\times10^5$ | 0.20 |
| DCF $M$=32, $K$=3 | 96.63 | $8.194\times10^4$ | 0.12 |
| SFCNN $N_\theta$=8, $M$=16 | $\sim$ 98.8 | - | - |
| RotDCF $N_\theta = 8$ | | | |
| $M$=16, $K$=14, $K_\alpha$=8 | 98.61 | $7.603\times10^5$ | 1.12 |
| $M$=16, $K$=5, $K_\alpha$=8 | 98.62 | $2.717\times10^5$ | 0.40 |
| $M$=16, $K$=5, $K_\alpha$=5 | 98.58 | $1.699\times10^5$ | 0.25 |
| $M$=16, $K$=3, $K_\alpha$=5 | 98.43 | $1.020\times10^5$ | 0.15 |
| $M$=8, $K$=14, $K_\alpha$=8 | 98.56 | $1.902\times10^5$ | 0.28 |
| $M$=8, $K$=5, $K_\alpha$=8 | 98.45 | $6.799\times10^4$ | 0.10 |
| $M$=8, $K$=5, $K_\alpha$=5 | 98.40 | $4.255\times10^4$ | 0.06 |
| $M$=8, $K$=3, $K_\alpha$=5 | 98.40 | $2.557\times10^4$ | 0.04 |

| rotMNIST Conv-6, $N_{tr} = 6K$ | | | |
|---|---|---|---|
| | Test Acc. | # param. | Ratio |
| CNN $M$=32 | 95.17 | | |
| DCF $M$=32, $K$=3 | 95.14 | | |
| SFCNN $N_\theta$=8, $M$=16 | $\sim$ 98.3 | - | - |
| RotDCF $N_\theta$=8 | | | |
| $M$=16, $K$=14, $K_\alpha$=8 | 98.00 | | |
| $M$=16, $K$=5, $K_\alpha$=5 | 97.95 | (same as left) | |
| $M$=8, $K$=5, $K_\alpha$=5 | 97.95 | | |
| $M$=8, $K$=3, $K_\alpha$=5 | 97.82 | | |

| CIFAR10 VGG-16, $N_{tr} = 10K$ | | | |
|---|---|---|---|
| CNN $M = 64$ | 78.40 | $2.732\times10^6$ | 1.00 |
| RotDCF, $N_\theta = 8$ | | | |
| $M$=32, $K$=3, $K_\alpha$=7 | 79.44 | $1.593\times10^6$ | 0.58 |
| $M$=32, $K$=3, $K_\alpha$=5 | 79.53 | $1.138\times10^6$ | 0.42 |

Table 3: Classification accuracy using non-equivariant CNN and rotation-equivariant ones on **rotMNIST** and **CIFAR10**. Baseline DCF (Qiu et al., 2018) and SFCNN (Weiler et al., 2017). "# param." is number of parameters in all convolutional layers, and "Ratio" indicates the proportion to the # param. of the regular CNN. Notice that the reduction from non-bases rotation-equivariant CNNs (the fair comparison case) can be even smaller, which is the factor of $\frac{KK_\alpha}{L^2 N_\theta}$, c.f. Section 2.3. The case of maximum $K$ and $K_\alpha$ ($K$=14, $K_\alpha$=8 in rotMNIST) is mathematically equivalent to using "full" filters without bases truncation.

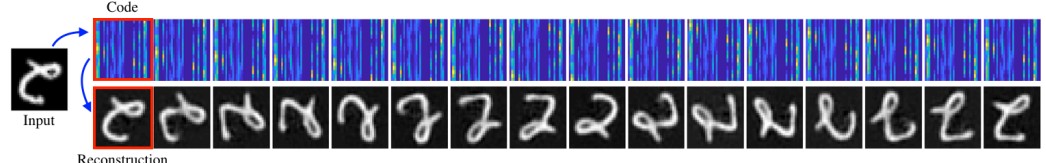

Figure 3: Codes and reconstructions of rotMNIST digits. (Top) A test image is encoded into a $16 \times 32$ array in the red box (the intermediate representation), and the code generates 16 copies by circulating the rows. (Bottom) Images reconstructed from the row-circulated codes above by the decoder.

a CNN with 6 convolutional layers (Table A.2), and we also compare with DCF (Qiu et al., 2018), which is non-rotation-equivariant but uses bases truncation, and SFCNN (Weiler et al., 2017), which is rotation-equivariant but does not use bases decompostion. The RotDCF net is made by replacing the regular convolutional layers with RotDCF layers, with $N_\theta$ many rotation-indexed channels and a reduced number of (unstructured) channels $M$. $K$ many $\psi_k$ bases and $K_\alpha$ many $\varphi_m$ are used. The classification accuracy is shown in Table 3. We see that RotDCF net obtains improved classification accuracy from CNN with significantly reduced number of parameters, e.g., with 12K training, Rot-DCF Net with $M$=8, $K$=3, $K_\alpha$=5 improves the test accuracy from $96.54$ to $98.40$ with less than $\frac{1}{20}$ many parameters of the CNN model and $\frac{1}{3}$ of the DCF model. The performance is also comparable to SFCNN which has significantly larger model complexity. The results are similar with reduced training size (5K) and on a shallower net with 3 conv layers, c.f. Table A.3.

The **CIFAR10** dataset consists of $32 \times 32$ colored images from 10 object classes (Krizhevsky, 2009), and we use 10,000 training and 50,000 testing samples. The network architecture is modified from VGG-16 net (Simonyan & Zisserman, 2014) (Table A.4). As shown in Table 3, RotDCF Net obtains better testing accuracy with reduced model size from the regular CNN baseline model. Throughout these experiments, the higher accuracy is due to the group equivarance and the lower model complexity is due to the bases decomposition.

**Transfer learning setting.** We train a regular CNN and a RotDCF Net on 10,000 up-right MNIST data samples, and directly test on 50,000 randomly rotated MNIST samples where the maximum rotation angle MaxRot=30 or 60 degrees (the "no-retrain" case). We also test after retraining the last two non-convolutional layers (the " fc-retrain" case). To visualize the importance of image regions which contribute to the classification accuracy, we adopt Class Activation Maps (CAM) (Zhou et al., 2016), and the network is modified accordingly by removing the last pooling layer in the net in Table A.1. and inserting a "gap" global averaging layer. The test accuracy are listed in Table 2, where the superiority of RotDCF Net is clearly shown in both the "no-retrain" and " fc-retrain" cases . The improved robustness of RotDCF Net is furtherly revealed by the CAM maps (Figure 2): the red-colored region is more stable for RotDCF Net even in the case with retraining.

## 4.2 IMAGE RECONSTRUCTION

To illustrate the explicit encoding of group actions in the RotDCF Net features, we train a convolutional auto-encoder on the rotated MNIST dataset, where encoder consists of stacked RotDCF layers, and the decoder consists of stacked transposed-convolutional layers (Table A.5). The encoder maps a $28 \times 28$ image into an array of $16 \times 32$, where the first dimension is the discretization of the rotation angles in $[0, 2\pi]$, and the second dimension is the unstructured channels. Due to the rotation equivariant relation, the "circulation" of the rows of the code array should correspond to the rotation of the image. This is verified in Figure 3: The top panel shows the code array produced

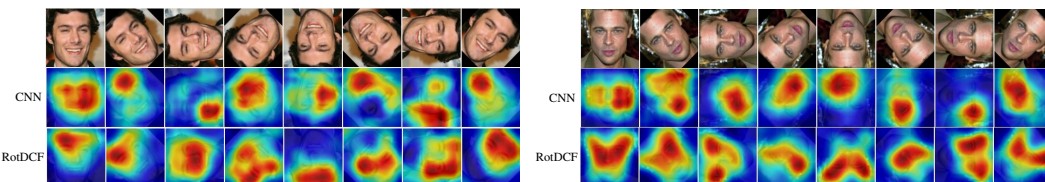

Figure 4: Example CAM maps for recognizing faces with in-plane rotations. The heatmap indicates the importance of different image regions used by respective models in defining a face, and a good CNN model is expected to select consistent face regions across the same-person images to determine the identity. Across different in-plane rotated copies, RotDCF chooses significantly more consistent discriminative regions than CNN, indicating more stable representations. In this experiment, we obtain 0.54% recognition accuracy using CNN (nearly random guess), and 97.04% accuracy using RotDCF with feature alignment, on known subjects.

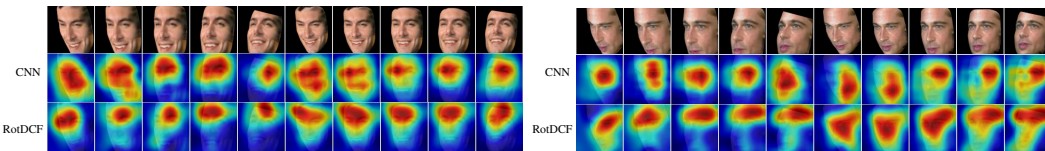

Figure 5: Synthesized faces from a testing image with -40$^o$ to 40$^o$ yaw, and -20$^o$ to 20$^o$ pitch, at a 10$^o$ interval.

Figure 6: Example CAM maps for recognizing faces with out-of-plane rotations. Across out-of-plane rotated copies, the discriminative regions chosen by RotDCF in describing a subject are more consistent, showing better representation stability than CNN. In this experiment, we obtain 80.79% recognition accuracy using CNN, and 89.66% using RotDCF, on known subjects.

from a testing image, and the 16 row-circulated copies of it. The bottom panel shows the output of the decoder fed with the codes in the top panel.

## 4.3 FACE RECOGNITION

As a real-world example, we test RotDCF on the Facescrub dataset (Ng & Winkler, 2014) containing over 100,000 face images of 530 people. A CNN and a RotDCF Net (Table A.6) are trained respectively using the gallery images from the 500 known subjects, which are preprocessed to be near-frontal and upright-only by aligning facial landmarks (Kazemi & Sullivan, 2014). See Appendix C for data preparation and training details. For the trained deep networks, we remove the last *softmax* layer, and then use the network outputs as deep features for faces, which is the typical way of using deep models for face verification and recognition to support both seen and unseen subjects (Parkhi et al., 2015). Using deep features generated by the trained networks, a probe image is then compared with the gallery faces whose identities are known and classified as that of the top match.

Under this gallery-probe face recognition setup, we obtain 94.10% and 96.92% accuracy for known and unknown subjects respectively using the CNN model; using RotDCF, the accuracies are 93.42% and 96.92%. Testing on unknown subjects are critical for validating the model representation power over unseen identities, and the reason for higher accuracy is simply due to the smaller number of classes. For both cases, RotDCF reports comparable performance as CNN, while the number of parameters in the RotDCF model is about one-fourth of the CNN model (see Appendix).

**In-plane rotation.** This experiment demonstrates the rotation-equivariance of the RotDCF features. We apply in-plane rotations at intervals of $\frac{\pi}{4}$ to the probe images (Figure 4), and let the original probe set be the new gallery, the rotated copies be the new probe set. In this setting, using the RotDCF model we obtain 97.04% and 97.58% recognition accuracy for known and unknown subjects respectively, after aligning the deep features by circular shifts (using the largest-magnitude $\alpha$ channel as reference). Notice that the model only sees upright faces. This is due to the rotation-equivariant property of the RotDCF Net, which means that the face representation is consistent regardless of its orientations after the group alignment. Lacking such properties, CNN obtains 0.54% and 5.05% recognition accuracies, which is close to random guess. We further compare CNN and RotDCF models via the CAM maps, which indicate the image regions relied by the CNN to make a classification prediction. In Figure 4, with regular CNN face regions used for prediction varies dramatically from image to image, and with RotDCF a significantly more consistent region mostly covering eyes and nose are selected across images, which explains its superior accuracy.

**Out-of-plane rotation.** To validate our theoretical result on representation stability under input deformations, we introduce out-of-plane rotations to the probe. Each probe image is fitted to a 3D face mesh, and rotated copies are rendered at the 10$^o$ intervals with -40$^o$ to 40$^o$ yaw, and -20$^o$ to 20$^o$ pitch, generating 45 synthesized faces in total (Figure 5). The synthesis faces at two poses (highlighted in red) are used as the new gallery, and all remaining synthesis faces form the new probe. The out-of-plane rotations here can be viewed as mild in-plane rotations plus additional variations, a situation frequently encountered in the real world. With this gallery-probe setup, the RotDCF model obtains 89.66% and 97.01% recognition accuracy for known and unknown subjects, and the accuracies are 80.79% and 89.97% with CNN. The CAM plots in Figure 6 also indicate that RotDCF Net chooses more consistent regions over CNN in describing a subject across different

poses. Since the out-of-plane rotations as in Figure 5 can be considered as in-plane rotations with additional variations, the superior performance of RotDCF is consistent with the theory in Section 3.

## 5  CONCLUSION AND DISCUSSION

This work introduces a decomposition of the filters in rotation-equivariant CNNs under joint steerable bases over space and rotations simultaneously, obtaining equivariant deep representations with significantly reduced model size and an implicit filter regularization. The group equivariant property and representation stability are proved theoretically. In experiments, RotDCF demonstrates improved recognition accuracy and better feature interpretability and stability on synthetic and real-world datasets involving object rotations, particularly in the transfer learning setting. To extend the work, implementation issues like parallelism efficiency and memory usage should be considered before the computational savings can be fully achieved. The framework should also extend to other groups and joint geometrical domains.

## ACKNOWLEDGEMENTS

Work partially supported by NSF, DoD, NIH and AFOSR.

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

SUPPLEMENTARY MATERIAL

## A  FLOPS CALCULATION IN SECTION 2 AND CHOICE OF HYPERPARAMETERS

When the input and output are both $W \times W$ in space, the forward pass in a regular convolutional layer needs $M_0'M_0W^2(1 + 2L^2) \sim 2L^2M_0'M_0W^2$ flops. (Each convolution with a $L \times L$ filter takes $2L^2W^2$, and there are $M_0'M_0$ convolution operations, plus that the summation over $\lambda'$ takes $W^2M_0'M_0$ flops.) In a rotation equivariant CNN without using bases, an convolutional layer would take $\sim 2M'MW^2L^2N_\theta^2$ flops.

In a RotDCF layer, the computation consists of three parts: (1) The inner-product with $\varphi_m$ bases takes $W^2M' \cdot 2N_\theta K_\alpha$ flops. (2) The spatial convolution with the $\psi_k$ bases takes $K_\alpha M'K \cdot 2L^2W^2$ flops. (3) The multiplication with $a_{\lambda',\lambda}(k,m)e^{-im(k)\alpha - im\alpha}$ and summation over $\lambda', k, m$ takes $MN_\theta(4KK_\alpha M' + 2W^2KK_\alpha M')$ flops (real-valued version). Putting together, the total is $2M'W^2K_\alpha(N_\theta + L^2K + MN_\theta K)$, and when $M$ is large, the third term dominates and it gives $2M'MW^2K_\alpha KN_\theta$. Thus the reduction by using bases-decomposed filters is again a factor of $\frac{K}{L^2} \cdot \frac{K_\alpha}{N_\theta}$, and the relative ratio with a regular CNN is about $\frac{M'M}{M_0'M_0} \cdot \frac{KK_\alpha N_\theta}{L^2}$.

The choice of hyperparameters $K$ and $K_\alpha$ can be interpreted as the "frequency" truncation both in $\mathbb{R}^2$ and in $S^1$, and it trades-off between filter regularization and filter expressiveness: The more severe the truncation, the less many trainable parameters, the smoother the filter (Figure A.1) and less overfitting, while the potential risk of underfitting is larger. This is revealed in Table 3: as $K$ and $K_\alpha$ are reduced, the accuracy decreases monotonically in the rotMNIST experiment. When $K_\alpha$ is reduced to 3 then underfitting becomes noticeable and accuracy drops even more (not reported). The parameter $N_\theta$ corresponds to spatial discretization along $S^1$ and, since we adopt a spectral approach, it does not affect the number of trainable parameters, which is determined by $K_\alpha$. In other words, the choice of $N_\theta$ is only constrained by memory storage.

## B  PROOFS IN SECTION 3

### B.1  THEOREM 3.1

*Proof of Theorem 3.1.* Note: The bases expansion under $\psi_{j_l,k}$ and $\varphi_m$ does not affect the form of convolutional layers, but only impose regularity of the filters, thus the group-equivariant property of RotDCF follows from the theorem.

Observe that the equivariant relation (6) is equivalent to that

$$T_\rho x^{(l)}[x^{(l-1)}] = x^{(l)}[T_\rho x^{(l-1)}] \tag{A.1}$$

for all $l$, where $T_\rho x^{(0)}$ means $D_\rho x^{(0)}$.

The sufficiency part: When $l = 1$, by (1),

$$T_\rho x^{(1)}[x^{(0)}](u, \alpha, \lambda) = \sigma\left(\sum_{\lambda'} \int_{\mathbb{R}^2} x^{(0)}(\rho_{u_0,t}u + v, \lambda')W_{\lambda',\lambda}^{(1)}(\Theta_{\alpha-t}v)dv + b^{(1)}(\lambda)\right),$$

$$x^{(1)}[D_\rho x^{(0)}](u, \alpha, \lambda) = \sigma\left(\sum_{\lambda'} \int_{\mathbb{R}^2} x^{(0)}(\rho_{u_0,t}(u + v), \lambda')W_{\lambda',\lambda}^{(1)}(\Theta_\alpha v)dv + b^{(1)}(\lambda)\right).$$

Since $\rho_{u_0,t}(u + v) = \rho_{u_0,t} + \Theta_t v$, we have that $T_\rho x^{(1)}[x^{(0)}] = x^{(1)}[D_\rho x^{(0)}]$ by a change of variable of $\Theta_t v \mapsto v$.

When $l > 1$, by (2),

$$T_\rho x^{(l)}[x^{(l-1)}](u, \alpha, \lambda) = \sigma\left(\sum_{\lambda'} \int_{\mathbb{R}^2} \int_{S^1} x^{(l-1)}(\rho_{u_0,t}u + v, \alpha', \lambda')W_{\lambda',\lambda}^{(l)}(\Theta_{\alpha-t}v, \alpha' - \alpha + t)dvd\alpha' + b^{(l)}(\lambda)\right),$$

$$x^{(l)}[T_\rho x^{(l-1)}](u, \alpha, \lambda) = \sigma\left(\sum_{\lambda'} \int_{\mathbb{R}^2} \int_{S^1} x^{(l-1)}(\rho_{u_0,t}(u + v), \alpha' - t, \lambda')W_{\lambda',\lambda}^{(l)}(\Theta_\alpha v, \alpha' - \alpha)dvd\alpha' + b^{(l)}(\lambda)\right).$$

Again, inserting $\rho_{u_0,t}(u+v) = \rho_{u_0,t} + \Theta_t v$, the claim follows by changing variables $\Theta_t v \mapsto v$ and $\alpha' - t \mapsto \alpha'$.

The necessity part: When $l = 1$, denote the general convolutional filter as $w^{(1)}(v; \lambda', \lambda, \alpha)$, and

$$x^{(1)}(u, \alpha, \lambda) = \sigma(\sum_{\lambda'} \int_{\mathbb{R}^2} x^{(0)}(u+v, \lambda') w^{(1)}(v; \lambda', \lambda, \alpha) dv + b^{(1)}(\lambda)).$$

Recall that

$$T_\rho x^{(1)}[x^{(0)}](u, \alpha, \lambda) = \sigma(\sum_{\lambda'} \int_{\mathbb{R}^2} x^{(0)}(\rho_{u_0,t} u + v, \lambda') w^{(1)}(v; \lambda', \lambda, \alpha - t) dv + b^{(1)}(\lambda)),$$

$$x^{(1)}[D_\rho x^{(0)}](u, \alpha, \lambda) = \sigma(\sum_{\lambda'} \int_{\mathbb{R}^2} x^{(0)}(\rho_{u_0,t}(u+v), \lambda') w^{(1)}(v; \lambda', \lambda, \alpha) dv + b^{(1)}(\lambda))$$

and then (A.1) with $l = 1$ holding for any $x^{(0)}$ implies that

$$\sum_{\lambda'} \int_{\mathbb{R}^2} x^{(0)}(\rho_{u_0,t} u + v, \lambda') w^{(1)}(v; \lambda', \lambda, \alpha - t) dv = \sum_{\lambda'} \int_{\mathbb{R}^2} x^{(0)}(\rho_{u_0,t}(u+v), \lambda') w^{(1)}(v; \lambda', \lambda, \alpha) dv.$$

By that $\rho_{u_0,t}(u+v) = \rho_{u_0,t} + \Theta_t v$, the above equality gives that

$$w^{(1)}(v; \lambda', \lambda, \alpha - t) = w^{(1)}(\Theta_t^{-1} v; \lambda', \lambda, \alpha), \quad \forall \alpha, t \in S^1.$$

Let $t = \alpha$, and $F_{\lambda',\lambda}(v) = w^{(1)}(v; \lambda', \lambda, 0)$, we have that

$$w^{(1)}(\Theta_\alpha^{-1} v; \lambda', \lambda, \alpha) = F_{\lambda',\lambda}(v),$$

and this gives that $w^{(1)}(v; \lambda', \lambda, \alpha) = F_{\lambda',\lambda}(\Theta_\alpha v)$. This proves that (1) is necessary.

When $l > 1$, consider the general convolutional filter as $w^{(l)}(v; \lambda', \lambda, \alpha', \alpha)$. Using a similar argument, (A.1) implies that

$$w^{(l)}(v; \lambda', \lambda, \alpha', \alpha - t) = w^{(l)}(\Theta_t^{-1} v; \lambda', \lambda, \alpha' + t, \alpha), \quad \forall \alpha, t \in S^1.$$

Let $t = \alpha$, and $F_{\lambda',\lambda}(v, \alpha') = w^{(l)}(v; \lambda', \lambda, \alpha', 0)$, then

$$F_{\lambda',\lambda}(v, \alpha') = w^{(l)}(\Theta_\alpha^{-1} v; \lambda', \lambda, \alpha' + \alpha, \alpha),$$

which gives that $w^{(l)}(v; \lambda', \lambda, \alpha', \alpha) = F_{\lambda',\lambda}(\Theta_\alpha v, \alpha' - \alpha)$, which proves that (2) is necessary.

$\square$

### B.2 Proposition 3.2

**Proposition B.1.** *For all $l$,*

$$B_{\lambda',\lambda}^{(l)}, C_{\lambda',\lambda}^{(l)}, 2^{j_l} D_{\lambda',\lambda}^{(l)} \leq \pi \|a_{\lambda',\lambda}^{(l)}\|_{FB},$$

*where*

$$B_{\lambda',\lambda}^{(l)} := \int_{\mathbb{R}^2} \int_{S^1} |W_{\lambda',\lambda}^{(l)}(v, \beta)| dv d\beta, \ l > 1, \quad B_{\lambda',\lambda}^{(1)} := \int_{\mathbb{R}^2} |W_{\lambda',\lambda}^{(1)}(v)| dv$$

$$C_{\lambda',\lambda}^{(l)} := \int_{\mathbb{R}^2} \int_{S^1} |v| |\nabla_v W_{\lambda',\lambda}^{(l)}(v, \beta)| dv d\beta, \ l > 1, \quad C_{\lambda',\lambda}^{(1)} := \int_{\mathbb{R}^2} |v| |\nabla_v W_{\lambda',\lambda}^{(1)}(v)| dv \quad \text{(A.2)}$$

$$D_{\lambda',\lambda}^{(l)} := \int_{\mathbb{R}^2} \int_{S^1} |\nabla_v W_{\lambda',\lambda}^{(l)}(v, \beta)| dv d\beta, \ l > 1, \quad D_{\lambda',\lambda}^{(1)} := \int_{\mathbb{R}^2} |\nabla_v W_{\lambda',\lambda}^{(1)}(v)| dv$$

*As a result,*

$$B_l, C_l, 2^{j_l} D_l \leq A_l,$$

*where*

$$B_l := \max\{\sup_\lambda \sum_{\lambda'=1}^{M_{l-1}} B^{(l)}_{\lambda',\lambda}, \sup_{\lambda'} \frac{M_{l-1}}{M_l} \sum_{\lambda=1}^{M_l} B^{(l)}_{\lambda',\lambda}\},$$

$$C_l := \max\{\sup_\lambda \sum_{\lambda'=1}^{M_{l-1}} C^{(l)}_{\lambda',\lambda}, \sup_{\lambda'} \frac{M_{l-1}}{M_l} \sum_{\lambda=1}^{M_l} C^{(l)}_{\lambda',\lambda}\}, \tag{A.3}$$

$$D_l := \max\{\sup_\lambda \sum_{\lambda'=1}^{M_{l-1}} D^{(l)}_{\lambda',\lambda}, \sup_{\lambda'} \frac{M_{l-1}}{M_l} \sum_{\lambda=1}^{M_l} D^{(l)}_{\lambda',\lambda}\},$$

*and thus (A2) implies that $B_l$, $C_l$, $2^{j_l} D_l \le 1$ for all l.*

*Proof of Proposition B.1.* The proof for the case of $l = 1$ is the same as Lemma 3.5 and Proposition 3.6 of Qiu et al. (2018). We reproduce it for completeness. When $l = 1$, it suffices to show that for $F(v) = \sum_k a_k \psi_k(v)$,

$$\int |F(v)|dv, \int |v||\nabla F(v)|dv, \int |\nabla F(v)|dv \le \pi(\sum_k \mu_k a_k^2)^{1/2}. \tag{A.4}$$

Rescaling to $\psi_{j_l,k}$ in $v$ leads to the desired inequality with the factor of $2^{j_l}$ for $D^{(l)}_{\lambda',\lambda}$. To prove (A.4), observe that $F$ is supported on the unit disk, and then $\|F\|_1$, $\int |v||\nabla F(v)|dv \le \|\nabla F\|_1 \le \sqrt{\pi}\|\nabla F\|_2$, where $\|\nabla F\|_2^2 = \pi \sum_k \mu_k a_k^2$ due to the orthogonality of $\psi_k$.

For $l > 1$, similarly, we only consider the rescaled filters supported on the unit disk in $v$. Let $F(v,\beta) = \sum_{k,m} a_{k,m}\psi_k(v)\varphi_m(\beta)$, $\beta \in S^1$, similarly as above, we have that

$$\int\int |F(v,\beta)|dvd\beta, \int\int |v||\nabla_v F(v,\beta)|dvd\beta \le \int\int |\nabla_v F(v,\beta)|dvd\beta$$

$$\le (\pi \int\int |\nabla_v F(v,\beta)|^2 dvd\beta)^{1/2}$$

recalling that $\int d\beta$ on $S^1$ has the normalization of $\frac{1}{2\pi}$. Again, $\int\int |\nabla_v F(v,\beta)|^2 dvd\beta = \pi \sum_{k,m} \mu_k a_{k,m}^2$ due to the orthogonality of $\psi_k$ and $\varphi_m$. This proves that

$$\int\int |\nabla_v F(v,\beta)|dvd\beta \le \pi(\sum_{k,m} \mu_k a_{k,m}^2)^{1/2},$$

which leads to the claim after a rescaling of $v$. $\qquad\qquad\square$

**Remark 1** (Remark to Proposition 3.2). *The proposition only needs $B_l$, defined in (A.3), to be less than 1 for all l, in a rotation-equivariant CNN, which is implied by (A2) by Proposition B.1.*

*Proof of Proposition 3.2.* The proof is similar to that of Proposition 3.1(a) of Qiu et al. (2018). Specifically, in (a), the argument is the same for $l = 1$, making use of the fact that

$$\int |w(\Theta_\alpha v)|dv = \int |w(v)|dv, \quad \forall \alpha \in S^1$$

and $\int_{S^1} d\alpha = 1$ due to the normalization of $\frac{1}{2\pi}$. For $l > 1$, the same technique proceeds with the new definition of $B^{(l)}_{\lambda',\lambda}$ as in (A.2) which involves the integration of $\int_{S^1}(\cdots)d\beta$. The detail is omitted.

To prove (b), we firstly verify that $x_0^{(l)}$ only depends on $\lambda$. When $l = 1$, $x_0^{(1)}(u, \alpha, \lambda) = \sigma(b^{(1)}(\lambda))$. Suppose that it holds for $(l-1)$, consider $l > 1$,

$$x_0^{(l)}(u, \alpha, \lambda) = \sigma(\sum_{\lambda'} \int_{S^1} \int_{\mathbb{R}^2} x_0^{(l-1)}(u+v, \alpha+\beta, \lambda') W_{\lambda',\lambda}^{(l)}(\Theta_\alpha v, \beta) dv d\beta + b^{(l)}(\lambda))$$

$$= \sigma(\sum_{\lambda'} x_0^{(l-1)}(\lambda') \int_{S^1} \int_{\mathbb{R}^2} W_{\lambda',\lambda}^{(l)}(\Theta_\alpha v, \beta) dv d\beta + b^{(l)}(\lambda))$$

$$= \sigma(\sum_{\lambda'} x_0^{(l-1)}(\lambda') \cdot \int_{S^1} \int_{\mathbb{R}^2} W_{\lambda',\lambda}^{(l)}(v', \beta) dv' d\beta + b^{(l)}(\lambda))$$

$$= x_0^{(l)}(\lambda).$$

Thus $x_0^{(l)}(u, \alpha, \lambda) = x_0^{(l)}(\lambda)$ for all $l$ (without index $\alpha$ for $l = 1$). The rest of the argument follows from that $\|x_c^{(l)}\| = \|x^{(l)} - x_0^{(l)}\| = \|x^{(l)}[x^{(l-1)}] - x^{(l)}[x_0^{(l-1)}]\| \le \|x^{(l-1)} - x_0^{(l-1)}\| = \|x_c^{(l-1)}\|$, where the inequality is by (a). □

## B.3 THEOREM 3.3

**Proposition B.2.** *In a RotDCF Net, under (A1), (A2), (A3), $c_1 = 4$, for any $l$,*

$$\|x^{(l)}[D_\rho \circ D_\tau x^{(0)}] - T_\rho \circ D_\tau x^{(l)}[x^{(0)}]\| \le 2c_1 l |\nabla \tau|_\infty \|x^{(0)}\|,$$

*where $D_\tau$ only acts on the space variable $u$ of $x^{(l)}$ similar to (9).*

*Proof of Proposition B.2.* We firstly establish that for all $l$,

$$\|x^{(l)}[T_\rho \circ D_\tau x^{(l-1)}] - T_\rho \circ D_\tau x^{(l)}[x^{(l-1)}]\| \le 2c_1 |\nabla \tau|_\infty \|x_c^{(l-1)}\|, \tag{A.5}$$

where $T_\rho$ is replaced by $D_\rho$ if applies to $x^{(0)}$ which does not have index $\alpha$. This is because that

$$x^{(l)}[T_\rho \circ D_\tau x^{(l-1)}] = T_\rho x^{(l)}[D_\tau x^{(l-1)}]$$

by Theorem 3.1, and that

$$\|T_\rho x^{(l)}[D_\tau x^{(l-1)}] - T_\rho \circ D_\tau x^{(l)}[x^{(l-1)}]\| = \|x^{(l)}[D_\tau x^{(l-1)}] - D_\tau x^{(l)}[x^{(l-1)}]\|$$

by the definition of $T_\rho$ (a rigid rotation in $u$, and a translation in $\alpha$). This term can be upper bounded by $c_1(B_l + C_l)|\nabla \tau|_\infty \|x_c^{(l-1)}\|$ (Lemma B.3), which leads to the desired bound under (A2) by Proposition B.1.

The rest of the proof is similar to that of Proposition 3.3 of Qiu et al. (2018): Write $x^{(l)}[D_\rho \circ D_\tau x^{(0)}] - T_\rho \circ D_\tau x^{(l)}[x^{(0)}]$ as the sum of the differences $x^{(l)}[x^{(j)}[D_\rho \circ D_\tau x^{(j-1)}]] - x^{(l)}[T_\rho \circ D_\tau x^{(j)}[x^{(j-1)}]]$ for $j = 1, \cdots, l$. The norm of the $j$-th term is bounded by $\|x^{(j)}[D_\rho \circ D_\tau x^{(j-1)}] - T_\rho \circ D_\tau x^{(j)}[x^{(j-1)}]\|$ due to Proposition 3.2 (a), which, by applying (A.5) together with Proposition 3.2 (b), can be bounded by $2c_1 |\nabla \tau|_\infty \|x^{(0)}\|$. Summing over $j$ gives the claim. □

*Proof of Theorem 3.3.* The proof is similar to that of Theorem 3.8 of Qiu et al. (2018). With the bound in Proposition B.2, it suffices to show that

$$\|T_\rho \circ D_\tau x^{(L)}[x^{(0)}] - T_\rho x^{(L)}[x^{(0)}]\| \le c_2 2^{-j_L} |\tau|_\infty \|x^{(0)}\|.$$

By the definition of $T_\rho$, the l.h.s. equals $\|D_\tau x^{(L)}[x^{(0)}] - x^{(L)}[x^{(0)}]\|$, which can be shown to be less than $c_2 |\tau|_\infty D_L \|x_c^{(l-1)}\|$ by extending the proof of Proposition 3.4 of Qiu et al. (2018), similar to the argument in proving Lemma B.3. The desired bound then follows by that $D_L \le 2^{-j_L} A_L \le 2^{-j_L}$ (Proposition B.1 and (A2)) and that $\|x_c^{(l-1)}\| \le \|x^{(0)}\|$ (Proposition 3.2 (b)). □

**Lemma B.3.** *In a rotation-equivariant CNN, $B_l$, $C_l$ defined as in (A.3), under (A1), (A3), for all $l > 0$, with $c_1 = 4$, $x_c^{(l)}$ as in Proposition 3.2,*

$$\|x^{(l)}[D_\tau x^{(l-1)}] - D_\tau x^{(l)}[x^{(l-1)}]\| \le c_1(B_l + C_l)|\nabla \tau|_\infty \|x_c^{(l-1)}\|.$$

| Conv-3 CNN-$M$ | Conv-3 RotDCF-$M$ |
|---|---|
| c5x5x1x$M$ ReLu ap2x2 | rc5x5x1x$M$ ReLu ap2x2 |
| c5x5x$M$x2$M$ ReLu ap2x2 | rc5x5x$N_\theta$x$M$x2$M$ ReLu ap2x2 |
| c5x5x2$M$x4$M$ ReLu ap2x2 | rc5x5x$N_\theta$x2$M$x4$M$ ReLu ap2x2 |
| fc64 ReLu fc10 softmax-loss | fc64 ReLu fc10 softmax-loss |

Table A.1: Conv-3 network architectures used in rotMNIST, $M = 32$, 16 or 8. c$L$x$L$x$M'$x$M$ stands for a convolutional layer of patch size $L$x$L$ and input (output) channel $M'$ ($M$). ap$L$x$L$ stands for $L$x$L$ average-pooling. In the RotDCF Net, rc$L$x$L$x$N_\theta$x$M'$x$M$ stands for a rotation-indexed convolutional layer, which includes $N_\theta$-times many number of filters except for the 1st rc layer (see Section 2). Batch-normalization layers (not shown) are used during training.

| Conv-6 CNN-$M$ | Conv-6 RotDCF-$M$ |
|---|---|
| c5x5x1x$M$ ReLu | rc5x5x1x$M$ ReLu |
| c5x5x$M$x1.5$M$ ReLu ap2x2 | rc5x5x$N_\theta$x$M$x1.5$M$ ReLu ap2x2 |
| c5x5x1.5$M$x2$M$ ReLu | rc5x5x$N_\theta$x1.5$M$x2$M$ ReLu |
| c5x5x2$M$x2$M$ ReLu ap2x2 | rc5x5x$N_\theta$x2$M$x2$M$ ReLu ap2x2 |
| c5x5x2$M$x3$M$ ReLu | rc5x5x$N_\theta$x2$M$x3$M$ ReLu |
| c5x5x3$M$x4$M$ ReLu ap2x2 | rc5x5x$N_\theta$x3$M$x4$M$ ReLu ap2x2 |
| fc64 ReLu fc10 softmax-loss | fc64 ReLu fc10 softmax-loss |

Table A.2: Conv-6 network architectures used in rotMNIST. Similar to Table A.1 with 3 more layers.

*Proof of Lemma B.3.* The proof is similar to that of Lemma 3.2 of Qiu et al. (2018). Specifically, when $l = 1$, the argument is the same, making use of the fact that $\int |w(\Theta_\alpha v)| dv = \int |w(v)| dv$, $\forall \alpha \in S^1$ and $\int_{S^1} d\alpha = 1$ due to the normalization of $\frac{1}{2\pi}$. When $l > 1$, the same technique applies by considering the joint integration of $\int_{\mathbb{R}^2} \int_{S^1} (\cdots) dv d\beta$ instead of just $dv$. The only difference is in using the new definitions of $B_{\lambda',\lambda}^{(l)}$ and $C_{\lambda',\lambda}^{(l)}$ for $l > 1$ as in (A.2), both of which involve the integration of $\int_{S^1} (\cdots) d\beta$. The detail is omitted. $\square$

# C EXPERIMENTAL DETAILS IN SECTION 4

## C.1 OBJECT RECOGNITION WITH ROTMNIST AND CIFAR10

In the experiments on rotMNIST dataset, the network architecture for Conv-3 and Conv-6 nets are shown in Table A.1 and Table A.2. Stochastic gradient descent (SGD) with momentum is used to train 100 epochs with decreasing learning rate from $10^{-2}$ to $10^{-4}$.

In the experiments on CIFAR10 dataset, the VGG16-like network architecture is shown in Table A.4. SGD with momentum is used to train 100 epochs with decreasing learning rate from $10^{-2}$ to $10^{-4}$.

| rotMNIST Conv-3, $N_{tr} = 10K$ | | | |
|---|---|---|---|
| | Test Acc. | # param. | Ratio |
| CNN $M$=32 | 95.67 | $2.570 \times 10^5$ | 1.00 |
| DCF $M$=32, $K$=5 | 95.58 | $5.158 \times 10^4$ | 0.20 |
| DCF $M$=32, $K$=3 | 95.69 | $3.104 \times 10^4$ | 0.12 |
| RotDCF $N_\theta = 8$ | | | |
| $M$=16, $K$=14, $K_\alpha$=8 | 97.86 | $2.871 \times 10^5$ | 1.12 |
| $M$=16, $K$=5, $K_\alpha$=8 | 97.81 | $1.026 \times 10^5$ | 0.40 |
| $M$=16, $K$=3, $K_\alpha$=8 | 97.77 | $6.160 \times 10^4$ | 0.24 |
| $M$=16, $K$=5, $K_\alpha$=5 | 97.96 | $6.419 \times 10^4$ | 0.25 |
| $M$=16, $K$=3, $K_\alpha$=5 | 97.95 | $3.856 \times 10^4$ | 0.15 |
| $M$=8, $K$=5, $K_\alpha$=5 | 97.81 | $1.610 \times 10^4$ | 0.06 |
| $M$=8, $K$=3, $K_\alpha$=5 | 97.59 | $9.680 \times 10^3$ | 0.04 |

| rotMNIST Conv-3, $N_{tr} = 5K$ | | | |
|---|---|---|---|
| | Test Acc. | # param. | Ratio |
| CNN $M$=32 | 94.04 | | |
| DCF $M$=32, $K$=3 | 94.08 | | |
| RotDCF $N_\theta$=8 | | | |
| $M$=16, $K$=3, $K_\alpha$=5 | 96.79 | (same as left) | |
| $M$=8, $K$=3, $K_\alpha$=5 | 96.53 | | |

Table A.3: Classification accuracy using non-equivariant CNN and rotation-equivariant CNNs on **rotMNIST** based on Conv-3, c.f. Table A.1. Similar results to the rotmnist experiments in Table 3 with a net of fewer (three) layers.

| VGG-16 CNN-$M$ | VGG-16 RotDCF-$M$ |
|---|---|
| c3x3x3x$M$ ReLu c3x3x$M$x$M$ ReLu c3x3x$M$x$M$ ReLu | rc3x3x3x$M$ ReLu rc3x3x$N_\theta$x$M$x$M$ ReLu rc3x3x$N_\theta$x$M$x$M$ ReLu |
| c3x3x$M$x$M$ ReLu c3x3x$M$x$M$ ReLu mp2x2 | rc3x3x$N_\theta$x$M$x$M$ ReLu rc3x3x$N_\theta$x$M$x$M$ ReLu mp2x2 |
| c3x3x$M$x2$M$ ReLu c3x3x2$M$x2$M$ ReLu | rc3x3x$N_\theta$x$M$x2$M$ ReLu rc3x3x$N_\theta$x2$M$x2$M$ ReLu |
| c3x3x2$M$x2$M$ ReLu c3x3x2$M$x2$M$ ReLu mp2x2 | rc3x3x$N_\theta$x2$M$x2$M$ ReLu rc3x3x$N_\theta$x2$M$x2$M$ ReLu mp2x2 |
| c3x3x2$M$x4$M$ ReLu c3x3x4$M$x4$M$ ReLu | rc3x3x$N_\theta$x2$M$x4$M$ ReLu rc3x3x$N_\theta$x4$M$x4$M$ ReLu |
| c3x3x4$M$x4$M$ ReLu c3x3x4$M$x4$M$ ReLu mp2x2 | rc3x3x$N_\theta$x4$M$x4$M$ ReLu rc3x3x$N_\theta$x4$M$x4$M$ ReLu mp2x2 |
| fc128 ReLu fc10 softmax-loss | fc128 ReLu fc10 softmax-loss |

Table A.4: VGG-16-like network architectures used in CIFAR10, $M = 64$ or 32. mp$L$x$L$ stands for $L$x$L$ max-pooling, and other notations similar to Table A.1.

| RotDCF ConvAE |
|---|
| rc5x5x1x8 ReLu ap2x2 |
| rc5x5x$N_\theta$x8x16 ReLu ap2x2 |
| rc5x5x$N_\theta$x16x32 ReLu ap2x2 |
| rc5x5x$N_\theta$x32x32 ReLu        $\leftarrow$ **Encoded representation** |
| fc128 ReLu ct5x5x128x16$N_\theta$ ReLu |
| ct5x5x16$N_\theta$x8$N_\theta$ (upsample 2x2) ReLu |
| ct5x5x8$N_\theta$x1 (upsample 2x2) Eucledian-loss |

Table A.5: Convolutional Auto-encoder network used in the image reconstruction experiment. Rot-DCF layers are used in the encoder network, with $N_\theta = 16$, $K = 5$, $K_\alpha = 5$ ($K = 8$, $K_\alpha = 15$ in the last RotDCF layer), and transposed-convolutional layers with upsampling are used in the decoder net. ap$L$x$L$ stands for $L$x$L$ average-pooling, and other notations similar to Table A.1.

## C.2    CONVOLUTIONAL AUTO-ENCODER FOR IMAGE RECONSTRUCTION

The network architecture is shown in Table A.5. The network is trained on 50,000 training samples, the training set is augmented by rotating each sample at 8 random angles, producing 400k training set. The network is trained for 10 epochs, where the learning rate decreases from $10^{-3}$ to $10^{-6}$.

## C.3    FACE RECOGNITION ON FACESCRUB

To facilitate the evaluation on both known and unknown subjects, we select the first 500 of the 530 identities as our training subjects. The remaining 30 subjects are used for validating out of sample performance, namely the unknown subjects. The experiment on unknown subjects is critical for face models to generate over unseen people. For both known and unknown subjects, we hold 10 images from each person as the probe images, and the remaining as the gallery images. The images are preprocessed by aligning facial landmarks using Kazemi & Sullivan (2014). and crop the aligned face images to $112 \times 112$ with color. Thus, both our CNN and RotDCF models are trained with near-frontal and upright-only face images.

The network architecture is shown in Table A.6. According to the formula in Section 2, the number of trainable parameters in the RotDCF Net is about $(\frac{1}{2})^2 \cdot \frac{K}{L^2} \cdot K_\alpha = \frac{1}{4}$ of that of the CNN.

## C.4    EFFECTS OF REGULARIZATION BY FILTER DECOMPOSITION

Sample trained filters in the reconstruction experiment are shown in Figure A.1, which shows that RotDCF obtains smoother trained filters as a result of the regularization applied by the bases truncation. In the reconstruction experiment, increasing the number of bases degrades the synthesis (Right

| CNN | RotDCF |
|---|---|
| c5x5x3x32 ReLu mp2x2 | rc5x5x3x16 ReLu mp2x2 |
| c5x5x32x64 ReLu mp2x2 | rc5x5x$N_\theta$x16x32 ReLu mp2x2 |
| c5x5x64x128 ReLu c5x5x128x128 ReLu mp2x2 | rc5x5x$N_\theta$x32x64 ReLu c5x5x64x64 ReLu mp2x2 |
| c5x5x128x256 ReLu c5x5x256x256 ReLu mp2x2 | rc5x5x$N_\theta$x64x128 ReLu c5x5x128x128 ReLu mp2x2 |
| c5x5x256x256 ReLu c5x5x256x256 ReLu gap13x13 | rc5x5x$N_\theta$x128x128 ReLu c5x5x128x128 ReLu gap13x13 |
| fc softmax | fc softmax |

Table A.6: Network architectures used in face experiments, notations as in Table A.1. In the RotDCF Net, $N_\theta = 8$, $K = 5$, $K_\alpha = 5$.

in Figure A.1). Such improved stability due to the truncated decomposition supports the analysis in Section 3.

Figure A.1: Trained filters in RotDCF (Left), and in ordinary rotation-equivariant CNNs (Middle) without truncated decomposition. (Right) Reconstructed digits from row-rotated codes where the AE net is trained with larger $K$ and $K_\alpha$, showing more overfitting than Figure 3 in the paper.

