# OpenReview forum: "RotDCF: Decomposition of Convolutional Filters for Rotation-Equivariant Deep Networks"
_ICLR.cc/2019/Conference_

### Official Review · AnonReviewer1 · 2018-11-02
**Interesting work with promising results with issues in experimental section**

**Rating:** 7
**Confidence:** 3

**Review:**

This work extends on [1] by constructing CNN filters using Fourier-Bessel (FB) bases for rotation equivariant networks. Additionally to [1] it extends the process with using SO(2) bases which allow to learn combination of rotated FB bases and ultimately achieve good performance with less parameters than standard CNN networks thanks to filter truncation.

In general, this work is well written and shows interesting results. However it lacks context with regards to other existing works. For example [2] also uses steerable filters for achieving rotation equivariance, however with different steerable bases (rotation harmonics instead of FB). It would be useful to clarify why FB bases are more appropriate for truncation, eventually providing empirical evidence (even though rotation harmonics would probably need more parameters). Authors mention [2], however disregard it due to computational complexity, which would be the same if the rotation harmonics bases were truncated as well.

Similarly, this work is not strong in evaluating against existing methods. It provides evaluation of the vanilla group equivariant networks in a similar configuration, but due to design choices in the training and test set, it is not possible to compare it against other algorithms and other steerable bases such as those from [2]. This degrades the results slightly as it does not allow to verify the baseline results from other works.

Additionally, it would be useful to provide an ablation study which would show how important the bases in SO(2) are important for the model accuracy. This would allow to compare the results against the [1] as the FB filters are steerable as well (Equation 4).

It is hard to reach a final rating for this submission. On one hand, it can be seen as an incremental improvement of [1] for a new domain of tasks, without a thorough comparison against existing methods. On the other hand, the paper is well written and the results look promising - evaluation verifies that the algorithm performs well in multiple tasks with a fraction of parameters.

Considering that authors plan to release the source code and that this conference aims for publishing novel ideas (and the goal of this work is to achieve rotation equivariance with less parameters, which hasn't been tackled before), I am inclined towards acceptance of this paper, even though the experiments can be significantly improved.

Unfortunately, I was not able to verify correctness of the provided proofs.

Additional minor issues:
* The paper does not specify what FB bases exactly are being used (such as in [table 1;1]), mainly it does not seem to specify the SO(2) bases.
* It would be useful to visualise K and K_\alpha in Figure 1.
* Citations, if not part of the sentence, should be in parentheses to improve readability (\citep for natbib).
* On page 8, end of first paragraph - wrong reference (see S.M.)
* L, in section 2.3 is not defined.

[1] Qiu, Qiang, et al. "DCFNet: Deep Neural Network with Decomposed Convolutional Filters.", ICML 2018
[2] Weiler, Maurice, et al. “Learning Steerable Filters for Rotation Equivariant CNNs.” CVPR 2018

---

> ### Comment · AnonReviewer1 · 2018-11-29
> **Response to authors comments**
>
> I would like to thank the authors for a thorough response and for clarification of a few points which I have missed. Additionally, I believe the added experiments (mainly Table A.3 and addition of the SFCNN results) further verify the generality of the proposed method.
>
> Even though the SFCNN provides slightly better results, authors correctly claim that it does with an increased model complexity. However with this I would like to encourage the authors, in line with R2's points #2 and #3 to provide more details regarding the wall clock speed of the current implementation of the algorithm versus the selected hyper-parameter selection. I believe this should not influence the decision of acceptance, as the complexity is proven theoretically, but it might shed light on the limitations of the current implementation (as mentioned in the conclusions).
>
> Few minor niggles:
> - The last sentence in paragraph 1.1 (added in later revisions) sounds relatively vague - how special?
> - When citing as part of a sentence, the citation should not be in parentheses (e.g. As shown in Darwin, 1859 (\citet); vs Evolution (Darwin, 1859) is a ... (\citep)). This is purely nit-picking but I believe it helps readability.

---

### Official Review · AnonReviewer3 · 2018-11-02
**Good paper:**

**Rating:** 7
**Confidence:** 4

**Review:**

Group-equivariant deep networks are used as a solution for rotation-equivariance in CNNs. However, they are computationally expensive as the number of filters increases by a factor proportional to the number of groups. Inspired by ideas of filter decomposition used in CNN model compression, the authors of this work instead propose to use steerable filters across space and rotation, as basis filters for achieving rotation-equivariance, which leads to computational efficiency.

The authors show improved accuracy and model compression with their proposed approach versus regular CNNs for several different tasks (MNIST, CIFAR, autoencoders and face recognition) for rotated and upright images.

Furthermore the authors theoretically prove and demonstrate empirically (via multiple experiments) the group equivariance property and the representational stability under input variations of their proposed architecture.

The work is novel and it solves an open research problem.

However, the one major criticism of the work is that in the experimental section, especially for the rotated MNIST and rotated face recognition tasks, the authors should compare the accuracy of their method with the latest state-of-the-art group-equivariant deep networks instead of just regular CNNs. This will help to truly understand whether their method is superior or comparable to the more computationally expensive group-equivariant networks that are specifically designed to handle rotations in terms of accuracy as well or not. The regular CCNs, which are not designed to handle rotations, are obviously bound to be inferior to their approach.

---

> ### Comment · AnonReviewer3 · 2018-11-27
> **Comment to Authors**
>
> I thank the authors for diligently addressing the points raised by the reviewers. I am satisfied with the authors revisions and it has made their paper more favorable for acceptance.

---

### Official Review · AnonReviewer2 · 2018-11-16
**Principled method with good results**

**Rating:** 7
**Confidence:** 2

**Review:**

Summary:
This paper combines the benefits of using joint steerable filters (using the SO(2) group) for designing rotation-equivariant CNNs with those of decomposing the filters (using Fourier-Bessel bases) for reducing the computational complexity. In addition, this leads to a compressed model and filter regularization. The authors give theoretical guarantees on the rotation equivariance and representation stability with respect to in and out of plane rotation. Empirical results show that the model attains better accuracy compared to CNNs and non-rotation-equivariant deep networks while using fewer parameters and also performs similarly to a rotation-equivariant model with much bigger capacity.

Pros:
- Theoretical guarantees, elegant approach
- Good empirical results compared to other models
- Desirable properties: rotation-equivariance, lower computational complexity, fewer parameters, robustness and guaranteed stability to deformations

Cons:
- Somewhat incremental technical novelty: combination of two previously published methods (Qiu et al. 2018 & Weiler et al. 2017)

Comments:
1. I believe the related work section can be improved by explaining more clearly the connection between your work and the cited ones and emphasizing the advantages and limitations of RotDCF compared to other methods In particular, a reader should be able to precisely understand what is the novelty of this work is and what were the technical challenges in combining previously published ideas (such as DCF and SFCNN)
2. How do you determine the truncation in practice? How robust is the method to this choice? What are the trade-offs between using a value that is too low or too high? It would be interesting to show how performance and complexity vary with this parameter
3. It would also be helpful to have a discussion on choosing the parameters K_{alpha} and N_{theta} and how this affects the performance, computational complexity and number of parameters. This would provide more intuition on the limits of this method and the types of data it can be used for
4. In section 2.3, it would be helpful to specify an estimated range for the parameter reduction from the non-bases rotation-equivariant CNN to RotDCF (similar to the ½ factor from RotDCF to regular CNN)
5. Eq. (4) seems to be missing the definition of R_{m,q}
6. The notation for the supplementary material was confusing at times. I would suggest using the more standard notation for the appendix which can also be a more specific reference (e.g. A.1, A.2, etc.)

---

### Author Response · Authors · 2018-11-11
**Thanks and response to the reviews**

We would like to thank the reviewers for reading our paper and giving valuable feedback. Please see below for our response, and the manuscript is also updated.

One common question raised by both reviewers is "to compare to the latest group-equivariant deep networks". In the revised version, we update the experiment on rotMNIST in Table 3, Page 6, including a comparison to the result in SFCNN [2] as suggested by R1. The performance of RotDCF is comparable to the more computationally expensive group-equivariant networks, even with bases truncation which reduces the model complexity.

To answer the other questions of R1:

- About Fourier Bessel (FB) bases: "what bases are being used" and "compare to rotation harmonics":

The FB bases used in the paper is the standard one (Abramowitz & Stegun 1964), and a formula is added in Sec. 2.2, new Eqn (4). It is the same FB bases used in [1], however, [1] considers usual CNN rather than rotation equivariant ones and does not exploit the steerable property of FB. Also note that the “joint steerable bases” of psi (FB for space) and phi (Fourier for orientation) are used together here to decompose the “joint convolution” Eqn. (1) (2) - the scheme is proved to be necessary for the group-equivariant property.

Compare to rotation harmonics [2], FB bases are orthonormal and the truncation of FB base has a frequency interpretation - it preserves the low-frequency components and discards the high-frequency ends. This is crucial for the regularization effect (Figure 3 and Figure A.1) of using truncated FB bases. Theoretically, the properties of FB bases are also key elements upon which the representation stability can be proved.

- "An ablation study and compare with [1]":

The rotMNIST experiment in Table 3 provides a comparison to DCF [1], which uses FB bases to decompose filters but is not designed to be rotation-equivariant: DCF gives similar performance to regular CNN, but inferior to RotDCF and [2]. This shows the importance of rotation-equivariant design in the network when handling input rotations.

The other minor points:
* K and Kalpha are added in the plot of Figure 1.
* Citations in parentheses now.
* Page 8: "S.M." refers to the Supplementary Material, defined on Page 3.
*  A definition of L added in Section 2.3.  L is the width/height of the convolutional filter, as visualized in Figure 1.

Thanks again for the reading!

---

### Author Response · Authors · 2018-11-21
**Thanks and response to R2**

We thank the reviewer for the reading, the supportive comments as well as the valuable suggestions.

R2 comments on the combination of the two previously published methods: DCF and SFCNN (Qiu et al. 2018 & Weiler et al. 2017).  In fact, one of our main focuses is to design a principled and elegant way to improve steerable-filter CNN like SFCNN, by exploiting the joint filter decomposition over the product geometry of R^2 x S^1 simultaneously, adopting the product steerable bases FB bases x Fourier bases. This design is highly non-trivial, and it achieves multiple desirable properties at the same time, including rotation-equivariance, lower computational complexity, fewer parameters, robustness and provable representation stability. The theoretical result is also new and differs from that in DCF - as now we need stability not just against input deformation but "module" the group action - and the theory is supported by experiments which involve image rotations. As suggested, we have added an explanation of the technical challenges in the related work section.

R2 also raises the question of "How to determine truncation and set parameters in practice, and how it affects performance, computational complexity and number of parameters?". At the moment, the choices of those hyper-parameters are mostly empirical, example values are given in Section 2.3 and in Table 3 & Table A.3. In the two tables, it is shown that the performance is not very sensitive to the choice of these parameters over a range, e.g., for rotMNIST, the accuracy maintained 98.40~98.60 for various choices of K and K_alpha down to 3 and 5. The parameter choice of K and K_alpha is interpreted as the "frequency" truncation (both in R^2 and in S^1), thus it trades-off between filter regularization and filter expressiveness. The choice of N_theta does not affect the number of parameters but affects storage. To better clarify all this, an explanation is added in Appendix A. The reduction factor from non-bases equivariant CNN to RotDCF is added in Section 2.3.

We have also corrected the suggested notation issues. Thanks again for reading!

---

### Public Comment · ~David_W._Romero1 · 2019-10-22
**Code online?**

Dear authors,

I am very interested in your line of work. While reading your paper I noticed that you state that "all codes will be publicly available" but I was not able to find any implementation online. Is there indeed an online implementation I was not able to find? If this is not the case, are you still planning to release your implementation?

Thank you very much for your time, attention and your nice work.

Best Regards,

David

---

### Meta-Review · Area_Chair1 · 2018-12-14
**Provably stable rotation equivariant networks**

**Confidence:** 5
**Recommendation:** Accept (Poster)

**Metareview:**

This paper builds on the recent DCFNet (Decomposed Convolutional Filters) architecture to incorporate rotation equivariance while preserving stability. The core idea is to decompose the trainable filters into a steerable representation and learn over a subset of the coefficients of that representation.
Reviewers all agreed that this is a solid contribution that advances research into group equivariant CNNs, bringing efficiency gains and stability guarantees, albeit these appear to be incremental with respect to the techniques developed in the DCFNet work. In summary, the AC believes this to be a valuable contribution and therefore recommends acceptance.